# The Protective Role of Curiosity Behaviors in Coping with Existential Vacuum

**DOI:** 10.3390/bs14050391

**Published:** 2024-05-06

**Authors:** Barbara Gawda, Adrian Korniluk

**Affiliations:** Department of Psychology of Emotion & Personality, University of Maria Curie-Sklodowska, 20-612 Lublin, Poland; adrian.korniluk@mail.umcs.pl

**Keywords:** curiosity, existential vacuum, psychosocial resources

## Abstract

An existential vacuum is experienced as a kind of crisis that one can cope with using his/her strengths. The related literature suggests that the important determinants of coping with existential emptiness include positive emotional and personality resources, and among these—Curiosity Behaviors. The purpose of this study is to describe the role of curiosity as an important factor in relation to emotional resources in individuals experiencing an existential vacuum. A survey was conducted using online tools (*n* = 484). The hypotheses about the direct and indirect relationships between Curiosity Behaviors and existential vacuum were tested using multiple regression analyses and mediations. The study involved a sample of adult participants representing the general population. The participants completed five questionnaires, the first one focusing on Curiosity Behaviors, such as seeking out novel and challenging experiences and engagement in activities that capture one’s attention (The Curiosity and Exploration Inventory), and the other tools being the Multidimensional Existential Meaning Scale, the Emotional Regulation Questionnaire, the Flourishing Scale, and the Depression Anxiety Stress Scale. The results indicate that Curiosity Behaviors characteristically negatively predict existential vacuum. In addition, other variables, such as Flourishing, seem to be useful in explaining the relationships between these factors. Flourishing along with Curiosity increases a sense of Meaning in Life. Our results present evidence showing the importance of Curiosity Behaviors in coping with the existential vacuum.

## 1. Introduction

### 1.1. Meaning in Life and Existential Vacuum

The extreme lack of a sense of Meaning in Life is characterized as an existential vacuum, a concept introduced into the literature by Frankl [1]. From a dimensional perspective, the existential vacuum is the extreme pole of the sense of Meaning in Life [2]. This deficit results in a lack of awareness of life purpose and, according to Frankl, manifests in a state of permanent meaninglessness and stagnation. In practical terms, existential emptiness appears to have particularly significant associations with occupational burnout [2,3], and with phenomena such as low spiritual well-being [4], or even decreased faith in science [5]. Therefore, it can be argued that existential emptiness is an important variable from the perspective of existential positive psychology, emphasizing the courage and responsibility needed to confront existential anxiety [6]. 

We based our study on the theoretical model of Existential Meaning in Life (MIL) proposed by George and Park [7]. MIL is defined as the extent to which one’s life is experienced as making sense, and as being directed and motivated by valued goals. MIL can be described by three aspects/subconstructs: Purpose, Comprehension, and Mattering. The subconstruct Purpose refers to the degree to which individuals experience their lives as being directed and motivated by valued life goals. Comprehension can be defined as the extent to which individuals perceive a sense of coherence and understanding regarding their lives. Mattering, according to George and Park ([7], p. 206) is defined as “the degree to which individuals feel that their existence is of significance, importance, and value in the world”. According to Costin and Vignoles, Mattering is “a feeling that one’s actions make a difference in the world”; it is a positive attachment and a sense of value the individual has towards their own life [8].

Exploring the potential positive aspects, we can draw conclusions about the protective roles of certain factors. Such factors are seen as psychosocial resources, i.e., personal and relational resources that are internally valuable or facilitate access to something of internal value [9,10]. 

### 1.2. Curiosity Behaviors and MIL

One of the fundamental resources providing protection against existential vacuum can be curiosity, defined as a state or a set of behaviors. Curiosity, as a psychological state, is triggered by an information gap, combined with a sense that closing the gap is possible and desirable, regardless of extrinsic rewards [11]. Research on curiosity as a state emphasizes its exploratory and absorptive nature (i.e., seeking new situations and the inclination to engage in those situations, respectively) [12,13]. Curiosity is described as a motivational determinant of exploratory behaviors [14]. It supports them in two ways: first, by evoking a strong desire to understand, and second, by fostering endurance and patience to delay gratification, the former pattern being comparable to impulsivity (with which it shares neural underpinnings) [15]. Curiosity can influence behavior both positively and negatively, partially explaining risky behaviors in adolescents [16]. It proves useful in explaining the process of learning in the context of group behaviors [17]. In some studies, curiosity is defined as a resource that protects against stress [18]. Based on a literature review, particularly related to positive psychology, it can be argued that curiosity is considered one of the most valuable character strengths [19,20]. Curiosity (as a personality trait) is negatively correlated with depression and positively associated with subjective psychological well-being [21]. It serves as a motivator for learning, influences decision-making, and plays a crucial role in healthy human development [22,23]. 

Significant relationships between curiosity and the sense of MIL can be found in some theories. Research indicates that curiosity is associated with a willingness for personal development and a need for a sense of MIL [24]. One approach to this construct is the model proposed by Reker and Wong [25,26]. This model postulates that personal MIL consists of two interconnected aspects: global MIL (an existential belief that life has a purpose and a coherence) and situational MIL (assigning personal meaning to specific experiences). This concept is similar to the theory proposed by Frankl [1], referring to general MIL and specific meanings in life. Experiences that require situational meaning can initiate and reinforce the search for a global sense of MIL in life. The exploratory element, which is emphasized here, is likely supported by curiosity as a trait. 

### 1.3. Flourishing and MIL

Flourishing is a significant factor in the context of psychotherapy and positive psychology [27], although there are challenges in the scientific understanding of the nature of Flourishing [28]. The concept of Flourishing is variously defined as “a combination of feeling good and functioning effectively” [29], “living the good life” [30], and a mixture of emotional well-being (hedonia) and positive functioning (eudaimonia) [31]. Flourishing is also characterized by both a high level of well-being and a low level of psychopathology [31], and/or complete human well-being [32]. The conceptualization of Flourishing proposed by Diener and colleagues [33] appears helpful in explaining the relationship between curiosity and the sense of MIL. The researchers suggest that Flourishing should be understood as success perceived by respondents in important areas such as relationships, self-esteem, purpose, and optimism. The Flourishing Scale, despite being presented as the “New Well-being Measures” [33], shows that Flourishing is not the same as well-being. Mental well-being is a broader concept and includes Flourishing, which refers to the eudaimonic perspective of individuals’ social and psychological functioning [34]. Flourishing includes good physical condition, positive relationships with other people, and the development of human potential in all areas of its functioning [30]. Unlike the concept of quality of life, the concept of thriving or good functioning, Flourishing is related to human activity and not to the conditions in which a person lives. According to Seligman [30], positive psychology should focus on enhancing human Flourishing by strengthening various dimensions of human well-being. Seligman stated that the concept of happiness does not reflect the nature of human Flourishing. According to Seligman, MIL is one of the components of Flourishing [30]. He proposes a model in which well-being is an operationalization of human Flourishing and distinguishes five of its dimensions: positive emotions, achievements, sense of purpose and meaning, relationships with others, and engagement. The connections with the activity were also articulated by Ryff and Singer [35] who proposed that Flourishing is the pursuit of perfection, related to the realization of human potential. In turn, according to Fredrickson and Losada [36], Flourishing means living in the optimal range of human functioning which is associated with goodness, generativity, personal development, and mental resilience. Furthermore, researchers found that individuals with high levels of Flourishing are characterized by high conscientiousness, extraversion, and low neuroticism, and that Flourishing shares more similarities with the eudaimonic dimension of well-being (which reflects self-realization and subjective cognitive-affective experiences) than the hedonistic one i.e., the presence of positive affects [37]. 

### 1.4. Emotional Regulation 

The concept of emotion regulation strategies is the theoretical framework developed by Gross [38] and was operationalized into a tool by Gross and John [39]. It is based on two dimensions: Cognitive Reappraisal and Expressive Suppression. Cognitive Reappraisal involves reinterpretation of a situation to decrease or increase emotional expression. The Expressive Suppression strategy directly impacts the emotional response by dampening one’s behaviors conveying the emotions. The proposed two-factor construct appears to have a sound up-to-date psychometric structure and the model was fitted to data [40,41].

The literature suggests a connection between curiosity and emotional states as well as emotional regulation. For instance, in the process of information-seeking, adolescents have been observed to experience both pleasure and pain, as well as emotions regulated at a metacognitive level [42]. It has been noted that interest-type epistemic curiosity (i.e., a desire for new information) corresponds to a fun, carefree, and optimistic approach to learning, whereas deprivation-type curiosity (i.e., aimed at reducing unpleasant experiences associated with the sense of lacking new knowledge) reflects greater caution and carefulness in the pursuit of knowledge [43]. 

### 1.5. Distress and MIL

Distress is defined as a construct consisting of symptoms of depression, anxiety, and stress [44,45]. The literature also points to the relationship between Distress and a sense of meaning, and that the relationship between Distress and a sense of meaning is not unidirectional. Distress can impact the sense of meaning and there is an influence of MIL on Distress too [46,47]. The authors generally agree that a high level of MIL in individuals is negatively related to Distress. This is in line with the theory by Frankl, who noted that depression derives from a lack of a sense of meaning [1]. It is noteworthy, however, that in the process of searching for MIL, which is linked to Distress, finding meaning is not strictly related to stress reduction [48]. Elements of Distress can therefore exist during the process of building meaning: i.e., during the search for, and the internalization of, meaningful experiences. The search for meaning is a basic state corresponding immanently to an experience of existential vacuum in individuals [49].

### 1.6. The Theoretical Model

We aim to examine the factors explaining the existential vacuum based on the George and Park Theory [7]. Research documents that there are consistent constructs with MIL, such as well-being, quality of life, and Flourishing [50]. All definitions of Flourishing show that this concept can be associated with MIL in various ways: Flourishing can impact MIL, and Flourishing can play an indirect role in the relationship between MIL and other variables such as cognitive resources, for instance, curiosity. Flourishing reflects generativity and personal development and is related to higher conscientiousness as well as activity. These patterns of Flourishing can interact with Curiosity Behaviors and can lead a person to understand or search for meaning more effectively. Curiosity may also coexist with the state of Flourishing [51] which is positively related to the sense of meaning at both the individual and societal level [52]. Thus, we hypothesized that Flourishing can be directly related to MIL and can also play an indirect role in the relationship between Curiosity and MIL.

Another important variable associated with MIL is Curiosity, which is considered to be positively related to subjective psychological well-being and motivation toward healthy development [21]. Curious people can self-develop better and show greater interest in existential problems than those who are not curious about themselves or the world. Curious people may be more willing to search for their MIL. Their broad mental horizons and rich interests may encourage them to search and discover MIL to a greater extent. In the process of searching for MIL, another factor, along with Curiosity, can be very important, i.e., the regulation of emotions. People searching for MIL can experience a variety of different emotions, often negative states such as dissatisfaction, anxiety, feelings of uncertainty, and so on. Emotion regulation skills are very helpful in solving existential dilemmas and dealing with negative emotions in the search for meaning. Thus, we took into consideration this important factor potentially related to Curiosity and MIL, i.e., emotional regulation strategies. 

Furthermore, we have often drawn attention to the fact that the search for MIL and the process of solving existential dilemmas is associated with unpleasant emotional experiences. Therefore, we included this variable, i.e., Distress, in our model and examined its relation to both Curiosity and MIL. We expect that there are negative relationships between Distress (understood as anxiety and depression) and MIL [48]. Distress, anxiety, and depression can block or inhibit the search for MIL. 

Our hypotheses are as follows:
**Hypothesis** **1.** *There is the positive predictive role of Curiosity Behaviors along with Flourishing and emotional regulation, and the negative role of Distress (Anxiety, Depression, and Stress) explaining existential vacuum (and the opposite i.e, Meaning in Life (and its aspects).*
**Hypothesis** **2.** *Flourishing mediates in the relationship between Curiosity Behaviors and existential emptiness (Meaning in Life and/or its aspects).*

Our assumptions are based on the findings that indicate an impact of emotional regulation and Curiosity on MIL. There are directional relationships between curiosity and emotion regulation, i.e., curiosity correlates positively with Emotional Regulation. Curiosity modifies the duration of emotional experiences [53]. There are findings that indicate that curiosity about aversive information can have positive cognitive effects. Acquiring aversive information in a safe environment makes a person feel in control of it (e.g., by differentiating such information, and learning to tolerate negative emotions) [54]. We consider that the state of curiosity is associated with feeling a spectrum of emotions, ranging from positive, including the pleasure of knowledge [43], to negative [16], even aversive emotions [55]. Curiosity exposes individuals to a variety of emotions, and, at the same time, reinforces the need to explore. It therefore seems to favor strategies based on reinterpretation of the emotional situation. 

It has also been shown that the cognitive reappraisal strategy is a significant positive predictor of MIL, and, conversely, the expression suppression strategy is a significant negative predictor of the sense of MIL [56]. In the process of searching for meaning, an emotional regulation strategy i.e., cognitive reappraisal seems to have a positive impact. It can modify one’s search (as well as assigning) for meaning. Cognitive Reappraisal can support Curiosity in searching for MIL by cognitively reinterpreting emotional experiences important for understanding the meaning of oneself and the world. In explaining the potential indirect role of emotional regulation strategies, we refer to the literature, which shows that there are negative factors, including negative emotional states, that decrease MIL and intensify the existential vacuum, and that emotional regulation competencies can modify the process of dealing with negative states while searching for solutions to existential problems [57]. In such states, cognitive reappraisal can be helpful and increase curiosity and MIL. The presented findings show there are cause-and-effect relationships between variables that justify mediation analysis. 

Our subsequent hypothesis is that:
**Hypothesis** **3.** *Emotional regulation strategies such as Cognitive Reappraisal and Expressive Suppression mediate in the relationship between Curiosity Behaviors and Meaning in Life (and its aspects).*

Our fourth hypothesis is related to the relationship between Curiosity Behaviors, Distress, and MIL. The relationships between them can vary. On the one hand, MIL can cause anxiety and distress, and on the other hand, high distress can increase MIL. The correlations between Distress and MIL, and between Curiosity and Distress are negative [48]. Curiosity was found to be negatively associated with depression, which means that higher Curiosity allows a person to better deal with depressive moods [21]. Likewise, research indicates that low curiosity (high need for cognition) can predict depression symptoms and anticipated anxiety [58]. Sometimes, curiosity can be an emotional negative state [31]. Distress can mediate the relationship between Curiosity and MIL. As we have shown Curiosity may be dependent on Distress and it may modify searching for MIL. Sometimes people experience greater happiness and lower Distress and are more curious and satisfied with their sense of meaning. Other times, people are unhappy, distressed, and less curious, and their sense of MIL is lower. During the search for MIL, various interactions between Distress and Curiosity can occur. These various interactions can even exist in the process of building meaning: i.e., during the search and the internalization of meaningful experiences [49]. 

Our last hypothesis is as follows:
**Hypothesis** **4.** *Distress (including Depression, Anxiety, and Stress) mediates in the relationship between Curiosity Behaviors and Meaning in Life (and its aspects).*

Based on the theoretical background and research findings indicating the relationship between Curiosity and MIL, Curiosity and emotion regulation [59], Curiosity and mental/social well-being [60], Curiosity and emotional Distress [61], and relationships between MIL and the listed variables [32,52,57,61], we developed a model of the potential relationships between these variables (Figure 1). Taking all these factors together, we aimed to examine what are their associations with MIL, to what extent they explain MIL, and what interactions between them exist that allow for an explanation of existential emptiness.

The current project is original in nature, as it systematizes the described relationships between several variables into a model based on the general framework of existential positive psychology. To date, it has been challenging to find reports focusing on such variables in a model within a single study. We believe that the results of this study will find application, for instance in therapeutic interventions.

This study aims to construct a model depicting the interaction between curiosity and existential vacuum (the state of decreased sense of Meaning in Life), along with mediating variables (such as Flourishing, emotions regulation, as well as Depression, Anxiety, and Stress). 

## 2. Materials and Methods

### 2.1. Participants 

The study involved a sample of 484 individuals (234 women and 247 men, 3 without gender indication) aged from 18 to 74 (M = 38.97, SD = 11.62). Participants’ educational attainments ranged from secondary (34.7%) to higher/university education (58.5%). 

### 2.2. Study Measures

#### 2.2.1. The Curiosity and Exploration Inventory (CEI-II)

The Curiosity and Exploration Inventory II, developed by Kashdan and colleagues [55] and adapted into Polish language by Kaczmarek, Bączkowski and Baran [21,62], is a self-report instrument comprising 10 items. It consists of two subscales: Stretching (willingness to seek out knowledge and new experiences) and Embracing (willingness to embrace the novel, uncertain, and unpredictable nature of everyday life). Participants rate items on a 5-point Likert scale from 1 (not at all true for me) to 5 (completely true for me). The Polish version in our study demonstrates a strong correlation between the two dimensions (r = 0.78, *p* < 0.001); thus, the two subscales were combined into one variable/measure of Curiosity Behaviors. The CEI-II has a high internal consistency (Cronbach’s α = 0.91).

#### 2.2.2. The Multidimensional Existential Meaning Scale (MEMS)

The Multidimensional Existential Meaning Scale by George and Park [63], adapted into the Polish language by Gerymski and Krok [64], is a self-report instrument comprising 9 items (reduced in the Polish version). It consists of three subscales: Comprehension (the degree to which individuals perceive a sense of coherence and understanding in their lives), Purpose (the extent to which individuals regard their lives as being focused on and motivated by valued life goals), and Mattering (the degree to which individuals perceive their existence in terms of significance, importance, and value in the world). Respondents rate items on a 7-point Likert scale that ranges from 1 (very strongly disagree) to 7 (very strongly agree). The MEMS in our study demonstrates a good internal consistency, Cronbach’s α = 0.92 for the entire scale and ranging between 0.77 and 0.90 for the subscales. 

#### 2.2.3. The Emotional Regulation Questionnaire (ERQ)

The Emotional Regulation Questionnaire, by Gross and John [39], adapted into the Polish language by Larionow [65], is a self-report instrument comprising 10 items. It consists of two subscales: Cognitive Reappraisal (assessing the cognitive change that involves construing a potentially emotion-eliciting situation in a way that alters its emotional impact) and Expressive Suppression (assesses the response modulation that involves inhibiting ongoing emotional expressive behavior). Participants rate items on a 7-point Likert scale from 1 (strongly disagree) to 7 (strongly agree). In our study, the ERQ demonstrated good internal consistency for the Cognitive Reappraisal (Cronbach’s α = 0.87) and for the Expressive Suppression (Cronbach’s α = 0.77) subscales.

#### 2.2.4. The Flourishing Scale (FS)

The Flourishing Scale by Diener et al. [33], experimentally adapted into Polish by Kaczmarek, is a self-report instrument comprising 8 items. It is unidimensional, and it assesses social–psychological prosperity, defined as a need for competence, relatedness, and self-acceptance. Participants rate items on a 7-point Likert scale from 1 (strong disagreement) to 7 (strong agreement). In our study, the Flourishing Scale demonstrated very good internal consistency with Cronbach’s α = 0.91.

#### 2.2.5. The Depression Anxiety Stress Scale-21 (DASS-21)

The Depression Anxiety Stress Scale by Lovibond and Lovibond [66], adapted into the Polish language by Makara-Studzińska et al. [67], is a self-report instrument comprising 21 items. It consists of three subscales, assessing the symptoms of Depression (1), Anxiety (2), and Stress (3). Participants rate items on a 4-point Likert scale ranging from 0 (“did not apply to me at all”) to 3 (“applied to me very much”). The internal consistency of the scale in our study is appropriate: Cronbach’s α = 0.96 for the total, as well as 0.90 for the Depression, 0.90 for the Anxiety, and 0.90 for the Stress subscales.

### 2.3. Procedure

The survey was conducted online from 2023 to 2024. The link to the survey was sent to the participants via the answeo.pl research panel. The study was conducted in accordance with the Declaration of Helsinki and approved by the Ethics Committee of Maria Curie-Sklodowska University (protocol code 17/2023).

### 2.4. Statistical Analyses

Descriptive statistics (means, standard deviations) were calculated. The distribution of the variables was normal, we checked it with the Kolmogorov–Smirnov test (see Table 1). First, the inter-correlations between all the variables were calculated to check the internal consistency of the variables in the sample (Appendix A, Table A1). Because of the high correlation between Depression and Anxiety, one variable was defined to include these two factors. At the next stage, we attempted to calculate one SEM model that included all our variables; however, the statistics of goodness of fit were not satisfying for one model. Thus, it was impossible to integrate the data into one model. Therefore, three multiple regression analyses were computed for the dependent variables of the MIL total, as well as Comprehension, Purpose, and Mattering, and for the independent variables of Curiosity Behaviors total, Cognitive Reappraisal, Expressive Suppression, Flourishing, Distress (encompassing Stress, Depression, and Anxiety due to the significant inter-correlations between Anxiety, Depression and Stress). Finally, mediation analyses (SPSS ver. 29, bootstrapping was used) were conducted, taking into account Flourishing, Cognitive Reappraisal, Expressive Suppression, and Distress (encompassing Anxiety, Depression, and Stress). 

## 3. Results

The socio-demographic characteristics are presented in Section 2.1 the participants section. Descriptive statistics for the self-report questionnaires used are presented below.

In order to verify the hypotheses, we decided to conduct four multiple regression analyses. The explained variables were both MIL conceptualized as a unidimensional factor, and in three dimensions i.e., Comprehension, Purpose, and Mattering. The independent variables were Curiosity Behaviors, Flourishing, Emotional Regulation Cognitive Reappraisal, Emotional Regulation Expressive Suppression, and total Distress.

Multiple regression analysis including Curiosity Behaviors and several variables explains about 68% of the variance of MIL. The Durbin–Watson statistic (1.92) shows no autocorrelation in the residuals. The significant predictors of Meaning in Life include Curiosity Behaviors, Flourishing, the emotional regulation strategies of Cognitive Reappraisal and Expressive Suppression. Higher Curiosity and Flourishing, as well as better emotional regulation strategy (Cognitive Reappraisal), are related to higher Meaning in Life i.e., lower existential vacuum (Table 2). Distress is not associated with Meaning in Life.

Similar multiple regression analyses were calculated for the components of Meaning in Life, i.e., Comprehension, Purpose, and Mattering.

The next multiple regression analysis, including Curiosity Behaviors and several variables, explains about 61% of the variance in Comprehension, as an aspect of MIL. The Durbin–Watson statistic (1.98) shows no autocorrelation in the residuals. The significant predictors of Comprehension include Curiosity Behaviors, Flourishing, and the emotional regulation strategy of Cognitive Reappraisal. Higher Curiosity and Flourishing, as well as better Cognitive Reappraisal as an aspect of emotional regulation, are related to higher Meaning in Life i.e., lower existential vacuum (Table 3). Higher Distress is also associated with existential Comprehension.

The third multiple regression analysis, including Curiosity Behaviors and several variables, explains about 54% of the variance of Purpose as an aspect of MIL. The Durbin–Watson statistic (1.82) shows no autocorrelation in the residuals. The significant predictors of Purpose include Curiosity Behaviors, Flourishing, and the emotional regulation strategy of Cognitive Reappraisal. Higher Curiosity and Flourishing as well as better Cognitive Reappraisal as an aspect of emotional regulation, are related to higher Purpose (an aspect of Meaning in Life), i.e., a lower existential vacuum (Table 4). 

The fourth multiple regression analysis including Curiosity Behaviors and other variables explains about 47% of the variance of Mattering as an aspect of MIL. The Durbin–Watson statistic (1.95) shows no autocorrelation in the residuals. The significant predictors of Mattering include Flourishing, and the emotional regulation strategies of Cognitive Reappraisal and Expressive Suppression. Higher Flourishing and better emotional regulation of Cognitive Reappraisal and Expressive Suppression are related to higher Mattering (Table 5). Higher Expressive Suppression is associated with lower Mattering. Curiosity behaviors and Distress are not associated with the Mattering dimension.

During the next step, three mediation analyses were performed to examine the relationship between Curiosity Behaviors and MIL. The mediators were Flourishing, Cognitive Reappraisal, and Distress, taking into account the results of the regression analyses. To perform these analyses, we used bootstrapping. We checked whether the variables met the assumptions for mediation. We found that Curiosity Behaviors correlate with MIL and with Flourishing, and that Flourishing correlates with MIL. 

The first mediation analysis shows that Curiosity Behaviors are positively related to Meaning in Life and Flourishing, that Flourishing is positively related to MIL (Figure 2), and that Flourishing positively mediates the relationship between Curiosity and MIL (indirect effect = 0.51, bootLLCI = 0.40, bootULCI = 0.62). The indirect effect is higher than the direct effect of Curiosity Behaviors on Meaning in Life, which is lower (0.16, bootLLCI = 0.08, bootULCI = 0.25). This relationship is named a cooperative suppression, which means that the mediator strengthens the effect of the independent variable on the dependent variable, even though the mediator has a greater impact on the dependent variable than the independent variable. Flourishing intensifies Curiosity Behaviors; both high Flourishing and Curiosity Behaviors correspond to higher Meaning in Life. 

The second mediation analysis shows the positive mediation role of Cognitive Reappraisal in the relationship between Curiosity and MIL (Figure 3). The indirect effect is smaller than the direct effect (indirect effect = 0.25, bootLLCI = 0.16, bootULCI = 0.36; direct = 0.42, bootLLCI = 0.30, bootULCI = 0.53), which means that Cognitive Reappraisal only partly mediates the relationship between Curiosity behaviors and MIL. Higher Curiosity Behaviors and better Cognitive Reappraisal lead to an increased sense of MIL. People who present curious behaviors and use cognitive reappraisal emotional regulation strategies can have a higher sense of meaning in life.

The third mediation analysis shows that Distress partly mediates the relationship between Curiosity and Comprehension as aspect of MIL (Figure 4). The indirect effect is significant; however, it is small, and smaller than the direct effect (indirect effect = 0.09, bootLLCI = 0.05, bootULCI = 0.11; direct effect = 0.48, bootLLCI = 0.18, bootULCI = 0.26). This means that the mediation effect of Distress in the relationship between Curiosity and Comprehension is rather small. The relationship is different than the previous ones. Although Curiosity Behaviors are positively related to Comprehension, Distress is negatively associated with Curiosity Behaviors and with Comprehension (higher Distress decreases Comprehension). This can mean that Distress leads to a decreased sense of Comprehension of MIL but does not impact the relationship between Curiosity and Comprehension. Individuals with higher Curiosity can present a lower level of Distress (Anxiety, Depression, and Stress), and individuals with higher Distress can be less curious and may experience an existential vacuum. 

## 4. Discussion

Although the protective role of various positive emotional resources has already been investigated [21,58,68,69], the present study is the first attempt to assess curiosity as a potential determinant of existential vacuum. The findings show evidence supporting or partly confirming a majority of the hypotheses formulated. The variables that were tested decrease the existential vacuum in varying degrees. 

Four independent multiple regression analyses were performed to assess whether, and to what extent, Curiosity Behaviors, Flourishing, Emotion Regulation Strategies, and Distress predict the self-perceived MIL. We identified a general pattern in the data, wherein variables associated with cognitive regulation strategies positively explained MIL. It is also important to emphasize the role of emotional factors of Expressive Suppression, and Distress, which was negatively related to Comprehension as an aspect of the sense of MIL. The results varied in explaining the relationships when the phenomenon was approached as a single dimension (one variable MIL) and as three separate dimensions (components of MIL). Our findings revealed intriguing determinants of the sense of MIL. Curiosity Behaviors, categorized as Stretching and Embracing, were found to significantly predict the sense of MIL [7,64,65]. The former types of behaviors, which involve an active approach to seeking out information in new situations, doing things that are complicated or challenging, looking for experiences that challenge the way of thinking about oneself and the world, looking for opportunities to test oneself and develop as a person, are significant predictors of MIL. Similarly, behaviors labelled with the term Embracing, i.e., enjoying the unpredictability of everyday life, searching for new experiences and information, doing things that are unusual, performing excitingly unpredictable tasks, and exploring unknown people, events and places, play a similar role as the Stretching-type activities, in the sense of MIL. The above Curiosity Behaviors explain Comprehension and Purpose as aspects of MIL. 

Curiosity was not found to be a significant factor in explaining the model of Mattering as an aspect of MIL. This may be linked to the fact that, while assessing the dimension of Mattering, the respondent is asked to evaluate their life on an absolute scale, and—compared to the other two dimensions which relate to the formulated goals—this task is of a far more abstract temporal nature [8]. It activates the processes involving memory retrieval and data processing, whereas Curiosity has a motivational dimension and is directly oriented towards the future [57]. 

Another important finding suggests that a positive role in increasing the sense of MIL is played by Curiosity Behaviors in combination with the factor of Flourishing. The present study has identified three aspects of the relationships between Curiosity Behaviors, Flourishing, and MIL that are original and valuable. Curiosity Behaviors, conceptualized as internal strengths or resources for development, lead to the formation of a positive self-image. Individuals with a curious disposition, through continuous exploration, can construct a positive view of the world that is explainable and well-structured [7]. They set clear goals, which they then achieve, and this subsequently contributes to the development of MIL. Unfortunately, curiosity also leads to negative states, and is aversive when unfulfilled [46], which may partially explain why the direct effect of Curiosity Behaviors on MIL is not always as consequential as the theory suggests. 

Curiosity Behaviors support individuals in constructing a healthy self-image. Two out of the three dimensions of MIL, Comprehension (perception of a sense of coherence, understanding life) and Purpose (perception of life as goal-oriented), seem to have a foundation in active engagement, which is fostered by curiosity [57,65]. This is illustrated by the following examples of behaviors from the CEI-II questionnaire [59]:“Everywhere I go‚ I am out looking for new things or experiences”;“I like to do things that are a little frightening”;“I frequently seek out opportunities to challenge myself and grow as a person”.

We found that the factors of Curiosity and Flourishing are related to Meaning in Life. Another study underscores a comparable relationship between Meaning in Life and mental well-being [70]. Psychological well-being is a central concept in positive psychology. The Flourishing Scale was developed to specifically operationalize the factor of well-being, by emphasizing issues associated with social relationships, a sense of competence, and purpose in life [33]. It can be generalized that the concept examined by the questionnaire is existential and may even be considered as an oxymoron of existentialism [71]. Some studies [72] distinguish two types of well-being: eudaimonic and hedonistic. The Flourishing Scale addresses the former. The two-factor model of well-being has a better fit to the data than the one-factor model. Therefore, Flourishing, as an operationalization of the eudaimonic, i.e., the cognitive dimension of well-being, supports the explanation of existential anxiety, along with other variables fitting into the category of cognitive strategies: Curiosity Behaviors and Cognitive Reappraisal. This observation may be of interest to therapists seeking to integrate elements of existential therapy with cognitive-behavioral approaches.

Our important finding of the present study suggests that Curiosity Behaviors and Cognitive Reappraisal are related to the sense of MIL. The emotional regulation strategy of Expressive Suppression, in general, was not found to explain the sense of MIL, but in general, its role was limited to explaining the Mattering aspect. Cognitive Reappraisal is a positive coping strategy for dealing with negative situations. In the ERQ questionnaire, this behavior is operationalized as a change in thinking aimed at improving an individual’s emotional state [39]. In a meta-analysis, it was observed that such a strategy can moderate the relationship between stressors and negative outcomes [73]. Expressive Suppression, as an alternative strategy, had a relatively smaller impact on the investigated models, especially considering the three-dimensional MIL approach. The above observation may support the therapeutic foundations of cognitive–behavioral or other related approaches. It particularly aligns with the Acceptance and Commitment Therapy, which aims for psychological flexibility associated with behavior change [74]. 

We found that Distress, included in the model with other variables, did not explain MIL except in the case of Comprehension; Distress negatively predicts Comprehension. We showed that Distress decreases the sense of Comprehension (an aspect of MIL), i.e., it promotes the development of an existential vacuum. Our results are consistent with the findings of Ostafin, Papenfuss, and Vervaeke, who showed an inverse association between MIL and psychological distress [74]. If the bidirectional relationship is considered, it appears that the Distress subscale can explain the interaction of the MIL and variables associated with mental crises such as distress or mental suffering. This finding supports the need to study multidirectional relationships in mental health model research. 

The present study has demonstrated both a direct and indirect relationship between Curiosity Behaviors and the development of the sense of MIL. The mediation analysis showed that Flourishing is a positive mediator in the relationship between Curiosity Behaviors and MIL. We found that Flourishing strengthens the effect of Curiosity Behaviors on MIL. Flourishing intensifies the impact of Curiosity Behaviors and contributes to higher MIL. Flourishing seems to be a very influential factor in MIL. It is related to the development of human potential in all areas of its functioning [35,75]. Thus, it can enhance Curiosity Behaviors. Flourishing indicates personal development [30], generativity, and mental resilience, according to Fredrickson and Losada [36], which is why its strengthened role in the relationship between Curiosity and MIL is invaluable. It is worth noting that some research also points to the important role of a bidirectional relationship between Flourishing and a sense of MIL [76].

The mediation analysis also showed that Cognitive Reappraisal is a positive mediator in the relationship between Curiosity Behaviors and MIL. Better Cognitive Reappraisal impacts Curiosity Behaviors, and increases the sense of MIL. People who are curious and use cognitive reappraisal strategies present a higher sense of meaning in life. The role of emotional regulation and emotional flexibility in mental and social well-being has been emphasized [57,62]. People who can successfully cope with negative emotional feelings and are able to regulate negative emotions present a higher sense of well-being and meaning in life [59,62,63]. Cognitive reappraisal impacts curiosity behaviors and leads to coping with existential problems. These capacities are of value not only in emotional regulation but also in solving some existential dilemmas.

The mediation analysis also showed that Distress is a negative and ineffective mediator in the relationship between Curiosity Behaviors and Comprehension (an aspect of MIL). Distress itself reduces a sense of Comprehension but does not impact the relationship between Curiosity and Comprehension. Curious individuals can present lower levels of Distress (i.e., anxiety, depression, and stress), just as individuals with higher distress can be less curious and they may experience an existential vacuum. Our findings are in line with another study on the within-person variability of curiosity and its relationship with flourishing and depression. Negative associations between curiosity lability and both life satisfaction and flourishing were observed [77]. Curiosity was higher on days of greater happiness and lower on days of greater depressed moods [77]. 

Our observations reflect the advantages of a model-based approach to understanding psychological resources in explaining complex states such as the sense of Meaning in Life. 

## 5. Conclusions

The most important protective factors against existential vacuum include Curiosity Behaviors associated with such emotional resources as Flourishing and Cognitive Reappraisal. Low emotional resources can effectively explain the existential vacuum. The factor that plays a negative role, i.e., increases existential vacuum, is distress, encompassing negative emotional states such as anxiety and depression. We showed that curiosity is related to Meaning in Life both directly and indirectly. In the second case, Flourishing and Cognitive Reappraisal play an important strengthening role. 

### The Significance of the Study

Given the results of our research, we see two types of significance.

The first is an application aspect. We reassure therapists, especially those working with approaches emphasizing the role of MIL, about the importance of therapeutic interventions aimed at strengthening curiosity. These therapeutic methods are described in other studies [78]; however, our contribution is to show the underlying mechanisms of these interventions. 

The second is the theoretical aspect. We present a unique theoretical and empirical model, particularly focusing on the role of curiosity, which is a universal interpersonal resource. The presented model of the relationship between curiosity and MIL is enriched with variables that have a strong basis in psychology; however, their associations have not often been of interest to researchers, particularly the dynamic interactions of Flourishing and MIL. Describing the model empirically, we provided quantitative evidence of the strength and direction of these relationships.

## 6. Limitations

The study has limitations. First, the sample is not representative of Poland. It is increasingly clear that cross-cultural frameworks are needed to consider the context in which Flourishing and Meaning in Life are measured. Although several scales we used have recently been validated in Poland and we believe that Flourishing and Meaning in Life are universal constructs, they can be cross-culturally contextualized. Future research directions can include an examination of cross-cultural contexts and different emotional and personality competencies, allowing for the establishment of a comprehensive model of individual resources important in understanding existential vacuum.

## Figures and Tables

**Figure 1 behavsci-14-00391-f001:**
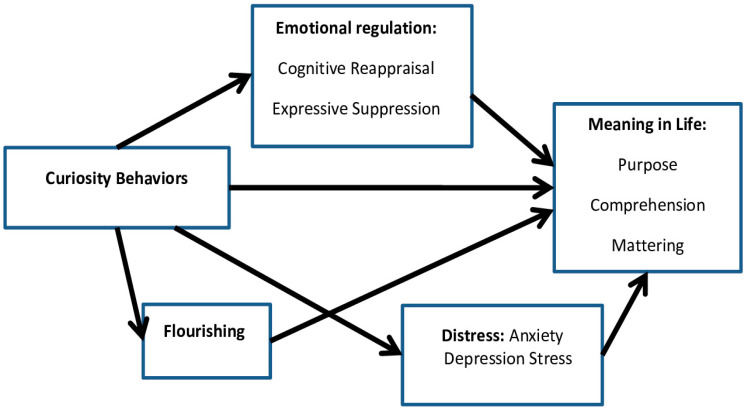
Model of determinants of Meaning in Life (MIL).

**Figure 2 behavsci-14-00391-f002:**
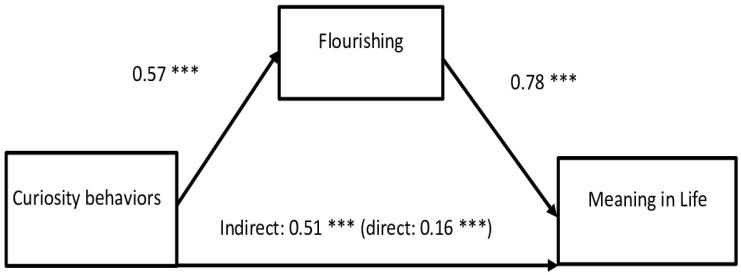
Mediation of Flourishing in the relationship between Curiosity Behaviors and MIL. Note: *** *p* < 0.001.

**Figure 3 behavsci-14-00391-f003:**
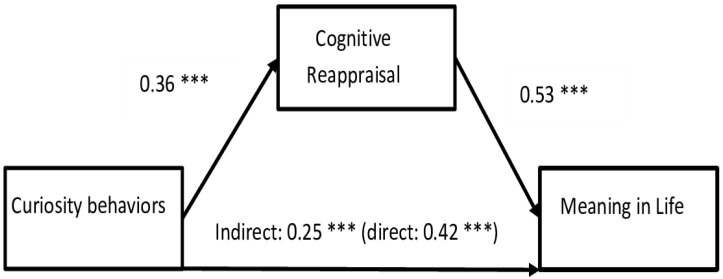
Mediation of Cognitive Reappraisal in the relationship between Curiosity Behaviors and MIL. Note: *** *p* < 0.001.

**Figure 4 behavsci-14-00391-f004:**
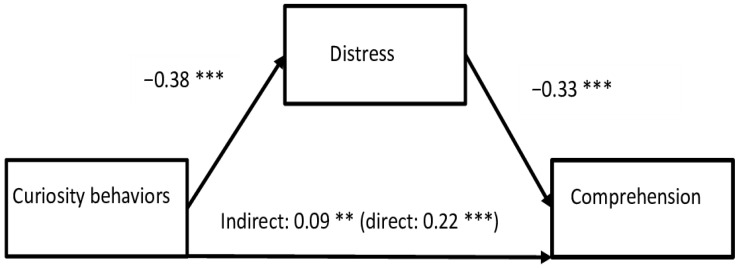
Mediation of Distress in the relationship between Curiosity Behaviors and Comprehension. Note: ** *p* < 0.01 *** *p* < 0.001.

**Table 1 behavsci-14-00391-t001:** Descriptive statistics (*n* = 484).

Variables	Min.	Max.	M	SD	Skew.	Kurt.	K-S *p*
MIL Meaning in Life	9.00	63.00	43.92	10.72	−0.653	0.303	0.06
MIL Comprehension	3.00	21.00	14.62	4.02	−0.725	0.377	0.12
MIL Purpose	3.00	21.00	15.65	3.94	−0.724	0.170	0.11
MIL Mattering	3.00	21.00	13.65	4.14	−0.476	0.126	0.10
ER Cognitive Reappraisal	6.00	42.00	27.56	632	−0.363	0.780	0.07
ER Expressive Suppression	4.00	28.00	16.54	4.65	−0.371	0.099	0.10
Flourishing	8.00	56.00	38.09	9.05	−0.453	0.172	0.08
Stretching Curiosity Beh	5.00	25.00	16.94	3.79	−0.507	0.590	0.10
Embracing Curiosity Beh	5.00	25.00	15.09	4.06	−0.450	0.000	0.09
Curiosity Behaviors	10.00	50.00	32.03	7.41	−0.510	0.488	0.08
Depression	7.00	28.00	14.23	5.82	0.567	−0.636	0.11
Anxiety	7.00	28.00	13.05	5.18	0.667	−0.367	0.14
Stress	7.00	28.00	14.65	5.16	0.364	−0.645	0.09
DASS Total (distress)	21.00	84.00	41.93	15.27	0.491	−0.577	0.09

Note: Abbreviations: MIL—Meaning in Life, ER—Emotional Regulation, Beh—Behaviors, K-S—Kolmogorov–Smirnov test.

**Table 2 behavsci-14-00391-t002:** Multiple regression analyses: predictors of Meaning in Life (*n* = 484).

Predictors	*B*	*SE*	*Beta*	*t*	*p*	*R*	*R* ^2^	*F*(5,477)
Cur Behaviors	0.124	0.044	0.086	3.617	<0.005	0.823	0.677	200.53 ***
Flourishing	0.796	0.042	0.672	18.965	<0.001			
ER Cogn Reappraisal	0.222	0.058	0.131	3.860	<0.001			
ER Expressive Suppression	−0.217	0.064	−0.094	−3.381	<0.001			
DASS Total (Distress)	−0.022	0.021	−0.032	−1.064	0.228			

Note: Abbreviations: ER—Emotional Regulation, Cogn—Cognitive, *SE*—standard error, *** *p* < 0.001.

**Table 3 behavsci-14-00391-t003:** Multiple regression analyses: predictors of Comprehension (*n* = 484).

Predictors	*B*	*SE*	*Beta*	*t*	*p*	*R*	*R* ^2^	*F*(5,477)
Cur Behaviors	0.061	0.018	0.111	3.355	<0.001	0.782	0.611	150.10 ***
Flourishing	0.279	0.001	0.627	16.120	<0.001			
ER Cogn Reappraisal	0.063	0.024	0.099	2.658	0.008			
ER Expressive Suppression	−0.028	0.026	−0.032	−1.042	0.298			
DASS Total (Distress)	−0.017	0.009	−0.065	−1.999	0.046			

Note: Abbreviations: ER—Emotional Regulation, Cogn—Cognitive, *SE*—standard error, *** *p* < 0.001.

**Table 4 behavsci-14-00391-t004:** Multiple regression analyses: predictors of Purpose (*n* = 484).

Predictors	*B*	*SE*	*Beta*	*t*	*p*	*R*	*R* ^2^	*F*(5,477)
Cur Behaviors	0.057	0.019	0.107	2.974	0.003	0.736	0.541	112.742 ***
Flourishing	0.266	0.018	0.610	14.427	<0.001			
ER Cogn Reappraisal	0.066	0.025	0.106	2.615	0.009			
ER Expressive Suppression	−0.007	0.028	−0.008	−0.234	0.815			
DASS Total (Distress)	−0.001	0.009	−0.005	−0.154	0.878			

Note: Abbreviations: ER—Emotional Regulation, Cogn—Cognitive, *SE*—standard error, *** *p* < 0.001.

**Table 5 behavsci-14-00391-t005:** Multiple regression analyses: predictors of Mattering (*n* = 484).

Predictors	*B*	*SE*	*Beta*	*t*	*p*	*R*	*R* ^2^	*F*(5,477)
Cur Behaviors	0.007	0.022	0.012	0.310	0.757	0.689	0.474	86.208 ***
Flourishing	0.252	0.021	0.549	12.144	<0.001			
ER Cogn Reappraisal	0.093	0.28	0.141	3.278	0.001			
ER Expr Suppression	−0.183	0.032	−0.205	−5.771	<0.001			
DASS Total (distress)	−0.004	0.010	−0.013	−0.350	0.727			

Note: Abbreviations: ER—Emotional Regulation, Cogn—Cognitive, *SE*—standard error, *** *p* < 0.001.

## Data Availability

The original contributions presented in the study are included in the article, further inquiries can be directed to the corresponding author.

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
