# Peer review of "The Protective Role of Curiosity Behaviors in Coping with Existential Vacuum"

_behavsci, 2024, doi:10.3390/bs14050391_

Round 1

Reviewer 1 Report

Comments and Suggestions for Authors

 The topic of this paper is ineresting but there are seeral issues to be addressed.

1. You wrote that “We have formulated hypotheses regarding the direct predic- 106 

tive role of Curiosity Behaviors in explaining existential vacuum (1) and the mediating  107 

role of Flourishing (2), Depression, Anxiety, and Stress  i.e. Distress  (3), emotion regula- 108 

tion (4) in the relationship between curiosity and existential emptiness.” Please add a section where you state your hypotheses.

2. The theoretical model should be placed at the end of the literature review.

3. There need to be section for hypothesis, where you provide the basis for proposing that flourishing is the mediator.

4. The discussion section must respond to the hy[potheses listed at the end of the introduction.

Comments on the Quality of English Language

Your use of language is fine.

Author Response

We would like to thank the Reviewer for all helpful comments to the manuscript. All suggestions have been considered. The required modifications have been implemented and are highlighted in yellow in the text of the manuscript and in the letter with responses.

Rev. 1

  1. You wrote that “We have formulated hypotheses regarding the direct predictive role of Curiosity Behaviors in explaining existential vacuum (1) and the mediating role of Flourishing (2), Depression, Anxiety, and Stress i.e. Distress (3), emotion regulation (4) in the relationship between curiosity and existential emptiness.” Please add a section where you state your hypotheses.

Res. : The section Hypotheses has been added.

1)         There is the direct predictive role of Curiosity Behaviors along with Flourishing, Emotional regulation, and Distress (Anxiety, Depression, and Stress) explaining existential vacuum/Meaning in Life (Meaning in Life and its aspects such as Purpose, Comprehension, and Mattering).

2)         Flourishing mediates in the relationship between Curiosity Behaviors and existential emptiness (Meaning in Life and/or its aspects, which aspect will be included in the analysis depends on the regression results and whether a variable met mediation assumptions).

3)         Emotional regulation strategies such as Cognitive Reappraisal and Expressive Suppression mediate in the relationship between Curiosity Behaviors and Meaning in Life (and/or its aspects).

4)         Distress (including Depression, Anxiety, and Stress) mediates in the relationship between Curiosity Behaviors and Meaning in Life (and/or its aspects).

  1. The theoretical model should be placed at the end of the literature review.

Res. We placed the theoretical model at the end of the literature review, we added the text and the figure 1.

The theoretical model

We aim to examine the factors explaining the existential vacuum. We based on the theoretical model of Existential Meaning in Life (MIL) proposed by George and Park [48]. MIL is defined as the extent to which one’s life is experienced as making sense, as being directed and motivated by valued goals. MIL can be described by three aspects/subconstructs: Purpose, Comprehension, and Mattering. The subconstruct Purpose refers to the degree to which individuals experience their lives as being directed and motivated by valued life goals. Comprehension can be defined as the extent to which individuals perceive a sense of coherence and understanding regarding their lives. Mattering, according to George and Park [48, p. 206] is defined as ”the degree to which individuals feel that their existence is of significance, importance, and value in the world”. According to Costin and Vignoles [49], Mattering is a feeling ”that one’s actions make a difference in the world”, it is a positive attachment and a sense of value the individual has towards their own life.

Research documents that there are theoretically consistent constructs with MIL such as well-being, quality of life, and flourishing [50]. The last one is variously defined as “a combination of feeling good and functioning effectively” [51], “living the good life” [32], and a mixture of emotional well-being (hedonia) and positive functioning (eudaimonia) [52]. Flourishing is also characterized by both a high level of well-being and low level of psychopathology [52], and/or complete human well-being [53]. According to Seligman [32] one of the components of Flourishing is Meaning in Life. Seligman [32] proposes a model in which well-being is an operationalization of human flourishing and distinguishes five of its dimensions: positive emotions, achievements, sense of purpose and meaning, relationships with others, and engagement. That is why we assume that there can be a direct and indirect relationship between Flourishing and MIL.

We consider other findings indicating that emotional resources such as emotional regulation competencies can impact MIL. Then, we take into account cognitive resources (i.e. curiosity) which are shown as positive factors playing a protective role against existential emptiness [54,55]. Literature documents there are also negative factors including negative emotional states such as depression and anxiety as well distress that decrease MIL and intensify the existential vacuum [56,57].

Based on the theoretical background and research findings showing the relationship between curiosity and flexibility of emotional reactions and emotion regulation [55], curiosity and mental/social well-being [58], curiosity and emotional distress [57], and that MIL is related to the ability to regulate emotions (for instance among cancer survivors) [54], mental well-being [59], emotional distress [56], and flourishing [27] we developed a model of the relationship between these variables (Fig. 1). Taking all these factors together, we aimed to check what are their associations with the existential vacuum, to what extent they explain MIL, and what interactions between them exist that allow explaining existential emptiness.

Figure 1. Model of determinants of Meaning in Life (red lines indicate potential mediation effects)

  1. There need to be section for hypothesis, where you provide the basis for proposing that flourishing is the mediator.

Res. This section was added, page Introduction. 

The Flourishing Scale, despite being presented as the "New Well-being Measures" [30], is not the same as well-being, it is broader concept and includes flourishing, which refers to the eudaimonic perspective of individuals' social and psychological functioning [31]. Flourishing includes good physical condition, positive relationships with other people, and the development of human potential in all areas of its functioning [32]. Unlike the concept of quality of life, the concept of thriving or good functioning Flourishing is related to human activity, and not to the conditions in which a person lives. According to Seligman [32], positive psychology should focus on enhancing human flourishing by strengthening various dimensions of human well-being. According to him, the concept of happiness does not reflect the nature of human Flourishing. Ryff and Singer [33] proposed that Flourishing is the pursuit of perfection, related to the realization of human potential. In turn, according to Fredrickson and Losada [34], flourishing means: living in the optimal range of human functioning which is associated with goodness, generativity, personal development, and mental resilience. Furthermore, researchers found that individuals with high level of flourishing are characterized by high conscientiousness, extraversion, and low neuroticism, and flourishing shares more similarities with the eudaimonic dimension of well-being than the hedonistic one [35].

Research documents that there are theoretically consistent constructs with MIL such as well-being, quality of life, and flourishing [50]. The last one is variously defined as “a combination of feeling good and functioning effectively” [51], “living the good life” [32], and a mixture of emotional well-being (hedonia) and positive functioning (eudaimonia) [52]. Flourishing is also characterized by both a high level of well-being and low level of psychopathology [52], and/or complete human well-being [53]. According to Seligman [32] one of the components of Flourishing is Meaning in Life. Seligman [32] proposes a model in which well-being is an operationalization of human flourishing and distinguishes five of its dimensions: positive emotions, achievements, sense of purpose and meaning, relationships with others, and engagement. That is why we assume that there can be a direct and indirect relationship between Flourishing and MIL.

  1. The discussion section must respond to the hypotheses listed at the end of the introduction.

Res.  The Discussion has been reorganized to respond to the hypotheses. The paragraphs were numbered according to the number of hypothesis.

Reviewer 2 Report

Comments and Suggestions for Authors

Thank you for this interesting article on the protective role of curiosity in coping with existential vacuum. I have some comments for the authors:

Abstract: 

- Lines 22-23: perhaps it would make sense to just say "... that Curiosity Behaviors characteristically negatively predict existential vacuum ..." - that way your next sentence is not needed.

- Lines 24-25: be more specific about how the variables are useful in explaining the relationship.

Introduction:

- Line 31. "V." is not needed.

- The second paragraph is a bit too long, can you sensibly split it into (at least) two parts?

- Can you write down your hypotheses in full verbatim?

Materials and Methods:

- Participants: what was the M and SD of age?

- It would be useful to have two more subsections at the end of the chapter: the Procedure and the Statistical Analysis (you already have some of the information written down, it would just be better if it were better structured).

Results:

- It would be useful to check the normality of the distribution with some other coefficients (Kolmogorov-Smirnov, Shapiro-Wilk ...).

- I also don't see any information on the internal consistency of the variables in your sample.

- I am not sure what happened to the numbering of the lines in Table 4.

- Figures 2 and 3: it would be useful to define what it means that the indirect effect is smaller than the direct effect.

Discussion:

- Lines 338-364: it might have been better if the text had not been written as numbered points. Numbers can also be added to "classic" text in the form of paragraphs.

- The limitations of your study and suggestions for further research in the field should be added to the chapter.

Author Response

We would like to thank the Reviewer for all helpful comments to the manuscript. All suggestions have been considered. The required modifications have been implemented and are highlighted in yellow in the text of the manuscript and in the letter with responses.

Rev. 2.

Abstract:

- Lines 22-23: perhaps it would make sense to just say "... that Curiosity Behaviors characteristically negatively predict existential vacuum ..." - that way your next sentence is not needed.

Res. We changed this sentence.

- Lines 24-25: be more specific about how the variables are useful in explaining the relationship.

Res. Explanation has been added.

Introduction:

- Line 31. "V." is not needed.

Res. We removed this.

- The second paragraph is a bit too long, can you sensibly split it into (at least) two parts?

Res. We split it.

- Can you write down your hypotheses in full verbatim?

Res. We wrote them.

Hypotheses. We have formulated the following hypotheses:

1)            There is the direct predictive role of Curiosity Behaviors along with Flourishing, Emotional regulation, and Distress (Anxiety, Depression, and Stress) explaining existential vacuum/Meaning in Life (Meaning in Life and its aspects such as Purpose, Comprehension, and Mattering).

2)            Flourishing mediates in the relationship between Curiosity Behaviors and existential emptiness (Meaning in Life and/or its aspects, which aspect will be included in the analysis depends on the regression results and whether a variable met mediation assumptions).

3)            Emotional regulation strategies such as Cognitive Reappraisal and Expressive Suppression mediate in the relationship between Curiosity Behaviors and Meaning in Life (and/or its aspects).

4)            Distress (including Depression, Anxiety, and Stress) mediates in the relationship between Curiosity Behaviors and Meaning in Life (and/or its aspects).

Materials and Methods:

- Participants: what was the M and SD of age?

Res. It has been added.

- It would be useful to have two more subsections at the end of the chapter: the Procedure and the Statistical Analysis (you already have some of the information written down, it would just be better if it were better structured).

Res. It has been done.

Results:

- It would be useful to check the normality of the distribution with some other coefficients (Kolmogorov-Smirnov, Shapiro-Wilk ...).

Res. The distribution has been checked, it is normal, we added the last column with p. for Kolmogorov-Smirnov test).

- I also don't see any information on the internal consistency of the variables in your sample.

Res.  It has been included. Appendix Table A1.

- I am not sure what happened to the numbering of the lines in Table 4.

Res. An error, It was corrected.

- Figures 2 and 3: it would be useful to define what it means that the indirect effect is smaller than the direct effect.

Res. It has been explained.

Fig. 2. The indirect effect is higher than direct effect of Curiosity Behaviors on Meaning in Life is lower (.16, bootLLCI =.08, bootULCI = .25). This relationship is named a cooperative suppression which means that the mediator strengthens the effect of the independent variable on the dependent variable, even mediator has greater impact on the dependent variable than the independent variable. Flourishing intensifies Curiosity Behaviors, both high Flourishing and Curiosity Behaviors correspond to higher Meaning in Life.

Figure 3. The second mediation analysis shows a positive mediation role of Cognitive Reap-praisal in the relationship between Curiosity and Meaning in Life (Fig. 3). The indirect effect is smaller than the direct effect (indirect effect = .25, bootLLCI =.16, bootULCI = .36; direct = .42, bootLLCI =.30, bootULCI = .53) which means that Cognitive Reappraisal only partly mediates in the relationship between Curiosity behaviors and Meaning in Life. Higher Curiosity Behaviors and better Cognitive Reappraisal lead to an increased sense of Meaning in Life. People who present curious behaviors and use cognitive reappraisal emotional regulation strategies can have a higher sense of meaning in life.

Figure 4. The indirect effect is significant, however small, and smaller than the direct effect (indirect effect = .09, bootLLCI =.05, bootULCI = .11; direct effect =.48, bootLLCI =.18, bootULCI = .26),). It means that a mediation effect of Distress in the relationship between Curiosity and Comprehension is rather small. The relationship is different than the previous ones. Although Curiosity Behaviors are positively related to Comprehension, Distress is negatively associated with Curiosity Behaviors and with Comprehension (higher Distress decreases Comprehension). It can mean that Distress leads to a decreased sense of Comprehension of Meaning in Life but does not impact the relationship between Curiosity and Comprehension.

Discussion:

- Lines 338-364: it might have been better if the text had not been written as numbered points. Numbers can also be added to "classic" text in the form of paragraphs.

Res. This has been modified.

- The limitations of your study and suggestions for further research in the field should be added to the chapter.

Res.  The Limitations have been added.

The study has limitations. First, the sample is not representative of Poland. It is increasingly clear that cross-cultural frameworks are needed to consider the context in which Flourishing and Meaning in Life are measured. Although several scales we used have recently been validated in Poland and we believe that Flourishing and Meaning in Life are universal constructs, they can be cross-culturally contextualized. Future research directions can include an examination of cross-cultural contexts, and different emotional and personality competencies allowing the establishment of a comprehensive model of individual resources important in understanding existential vacuum.

Reviewer 3 Report

Comments and Suggestions for Authors

I am really very thankful for getting the opportunity to read and review this article.

The article is very interesting with lots of theoretical and practical outcomes.

I would like to point out two gaps which should be taking into concideration before final publication to make the article more communicative to the readers who might be not highly specialised in positive psychology.

Firstly  - the concept of Flourishing - it is refered properly however it is not defined in detailes. Please be more specific and show this concept as more adequate as well-being in your research project. In the introduction as well as in the disccusion Flourishing and well-being are presented as synonimic concepts . However Flousishing is specific tear/theory/concept and is operationalised by specific instrument.

Secondly - the aim of the study. It is not clear why Depression, Anxiety, and Stress was assumed to be the mediating factor in the relationship between curiosity and existential emptiness.

Theoretical background for this assumption should be more clearly expressed due to the  fact that Depression, in reference to the Frankl's theory, results from the loss of the ability to make meaning.

If these theoretical assumptions and background were described in more detailed way the statictics of analysed regression and three mediation models would be understood and presented with  sufficient precision and accuracy.

It seems that such theoretical supplementation will not pose a problem for the authors (it is rather the minor revision) hoewever it is fundamental for improving the text.

Author Response

We would like to thank the Reviewer for all helpful comments to the manuscript. All suggestions have been considered. The required modifications have been implemented and are highlighted in yellow in the text of the manuscript and in the letter with responses.

Rev 3.

Firstly  - the concept of Flourishing - it is refered properly however it is not defined in detailes. Please be more specific and show this concept as more adequate as well-being in your research project. In the introduction as well as in the disccusion Flourishing and well-being are presented as synonimic concepts . However Flourishing is specific tear/theory/concept and is operationalised by specific instrument.

Res. This explanation has been provided in the Introduction as well in the Discussion.

In the Introduction

The researchers suggest that Flourishing should be understood as success perceived by the respondent in important areas such as relationships, self-esteem, purpose, and optimism. The Flourishing Scale, despite being presented as the "New Well-being Measures" [30], is not the same as well-being, it is broader concept and includes flourishing, which refers to the eudaimonic perspective of individuals' social and psychological functioning [31]. Flourishing includes good physical condition, positive relationships with other people, and the development of human potential in all areas of its functioning [32]. Unlike the concept of quality of life, the concept of thriving or good functioning Flourishing is related to human activity, and not to the conditions in which a person lives. According to Seligman [32], positive psychology should focus on enhancing human flourishing by strengthening various dimensions of human well-being. According to him, the concept of happiness does not reflect the nature of human Flourishing. Ryff and Singer [33] proposed that Flourishing is the pursuit of perfection, related to the realization of human potential. In turn, according to Fredrickson and Losada [34], flourishing means: living in the optimal range of human functioning which is associated with goodness, generativity, personal development, and mental resilience. Furthermore, researchers found that individuals with high level of flourishing are characterized by high conscientiousness, extraversion, and low neuroticism, and flourishing shares more similarities with the eudaimonic dimension of well-being than the hedonistic one [35].

Research documents that there are theoretically consistent constructs with MIL such as well-being, quality of life, and flourishing [50]. The last one is variously defined as “a combination of feeling good and functioning effectively” [51], “living the good life” [32], and a mixture of emotional well-being (hedonia) and positive functioning (eudaimonia) [52]. Flourishing is also characterized by both a high level of well-being and low level of psychopathology [52], and/or complete human well-being [53]. According to Seligman [32] one of the components of Flourishing is Meaning in Life. Seligman [32] proposes a model in which well-being is an operationalization of human flourishing and distinguishes five of its dimensions: positive emotions, achievements, sense of purpose and meaning, relationships with others, and engagement. That is why we assume that there can be a direct and indirect relationship between Flourishing and MIL.

In the Discussion

The mediation analysis showed that Flourishing is a positive mediator in the relationship between Curiosity Behaviors and Meaning in Life. We found that Flourishing strengthens the effect of Curiosity Behaviors on Meaning in Life. Flourishing intensifies the impact of Curiosity Behaviors and contributes higher Meaning in Life. Flourishing seems to be a very influential factor in Meaning in Life. It is related to the development of human potential in all areas of its functioning [33, 73]. Thus, it can enhance Curiosity Behaviors. Flourishing indicates personal development [32], generativity, and mental resilience according to Fredrickson and Losada [34] that is why its strengthened role in the relationship between Curiosity and Meaning in Life is invaluable. It is worth noting that some research also points to the important role of a bidirectional relationship between Flourishing and a sense of Meaning in Life [74].

Secondly - the aim of the study. It is not clear why Depression, Anxiety, and Stress was assumed to be the mediating factor in the relationship between curiosity and existential emptiness. Theoretical background for this assumption should be more clearly expressed due to the fact that Depression, in reference to the Frankl's theory, results from the loss of the ability to make meaning.

Res. The theoretical  background has been provided, and the relationship between Meaning in Life and Depression, anxiety, and Distress.

Distress. The relationship of distress and sense of meaning (MIL) is also not one-directional. The literature points to the influence of Meaning in Life on distress [42,43]. The authors generally agree that a high level of Meaning in Life in individuals is negatively related to distress. This is in line with the theory by Frankl, who noted that depression derives from a lack of a sense of meaning [1]. It is noteworthy, however, that in the process of searching for meaning in life which is linked to distress, finding meaning is not strictly related to stress reduction [44]. Elements of distress therefore exist during the process of building meaning: i.e., during the searching and the internalization of meaning experiences. The search for meaning is a basic state corresponding immanently with an experience of existential vacuum in individuals [45].

Distress can mediate the relationship between curiosity and meaning in life. Some-times, curiosity can be an emotional negative state. When it is unfulfilled, brain regions responsible for aversive states are activated [46]. Some studies have shown the neural and behavioral similarities between curiosity and impulsivity, perceived as a "negative state" [14]. Despite its dimensionality, curiosity plays a supportive role when confronted with distress. Curiosity is negatively associated with depression [20]. A low need for cognition (which is the opposite of curiosity) can predict depression symptoms and anticipated anxiety [47]. Thus, curiosity can be a protective resource.

Later in the Introduction …Literature documents there are also negative factors including negative emotional states such as depression and anxiety as well distress that decrease MIL and intensify the existential vacuum [56,57].

If these theoretical assumptions and background were described in more detailed way the statictics of analysed regression and three mediation models would be understood and presented with sufficient precision and accuracy.

Res. We improved it and clarified.

Round 2

Reviewer 1 Report

Comments and Suggestions for Authors

I thank the authors for revisions, but the hypothesis section is still not clear. Furthermore, the word "assume" is not correct. 

Please try to show readers a crystal logic for your hypotheses, not just describing the results from prior studies.

Comments on the Quality of English Language

it is ok.

Author Response

Res. We thank very much the Reviewer for all helpful comments to the manuscript. All suggestions have been considered and the required modifications have been implemented and are highlighted in yellow in the text of the manuscript and in the letter with responses.

Rev. 1

I thank the authors for revisions, but the hypothesis section is still not clear. Furthermore, the word "assume" is not correct.

Res. A word ‘assume’ was replaced by a word ‘hypothesize”

Please try to show readers a crystal logic for your hypotheses, not just describing the results from prior studies.

Response: We re-wrote/modified this section and we reorganized this section. We hope it is clear now.

We also modified the Figure 1 and we added the lines indicated the direct and indirect relationships between variables.

The section is as follows.

The theoretical model

We aim to examine the factors explaining the existential vacuum. We based our study on the theoretical model of Existential Meaning in Life (MIL) proposed by George and Park [46]. MIL is defined as the extent to which one’s life is experienced as making sense, as being directed and motivated by valued goals. MIL can be described by three aspects/subconstructs: Purpose, Comprehension, and Mattering. The subconstruct Purpose refers to the degree to which individuals experience their lives as being directed and motivated by valued life goals. Comprehension can be defined as the extent to which individuals perceive a sense of coherence and understanding regarding their lives. Mattering, according to George and Park [46, p. 206] is defined as ”the degree to which individuals feel that their existence is of significance, importance, and value in the world”. According to Costin and Vignoles, Mattering is “a feeling that one’s actions make a difference in the world”, it is a positive attachment and a sense of value the individual has towards their own life [47].

Research documents that there are consistent constructs with MIL, such as well-being, quality of life, and flourishing [48]. The concept of Flourishing is variously defined as “a combination of feeling good and functioning effectively” [49], “living the good life” [32], and a mixture of emotional well-being (hedonia) and positive functioning (eudaimonia) [50]. Flourishing is also characterized by both a high level of well-being and a low level of psychopathology [50], and/or complete human well-being [51]. According to Seligman, MIL is one of the components of Flourishing [32]. He proposes a model in which well-being is an operationalization of human flourishing and distinguishes five of its dimensions: positive emotions, achievements, sense of purpose and meaning, relationships with others, and engagement. All these definitions of Flourishing show that this concept can be associated with MIL in various ways: Flourishing can impact MIL, and Flourishing can play an indirect role in the relationship between MIL and other variables such as cognitive resources, for instance, curiosity. Flourishing reflects generativity and personal development and is related to higher conscientiousness as well as activity. These patterns of Flourishing can interact with Curiosity Behaviors and can lead a person to understand or search for meaning more effectively. Thus, we hypothesized that Flourishing can be directly related to MIL and can also play an indirect role in the relationship between Curiosity and MIL.

Another important variable explaining MIL is Curiosity, which is considered to be positively related to subjective psychological well-being and motivation toward healthy development [20]. Curious people can self-develop better and show greater interest in existential problems than those who are not curious about themselves or the world. Curious people may be more willing to search for their MIL. Their broad mental horizons and rich interests may encourage them to search and discover MIL to a greater extent. In the process of searching for MIL another factor along with Curiosity can be very important, i.e. regulation of emotions. People searching for MIL can experience a variety of different emotions, often negative states such as dissatisfaction, anxiety, feelings of uncertainty, and so on. Emotion regulation skills are very helpful in solving existential dilemmas and dealing with negative emotions in the search for meaning. Thus, we took into consideration this important factor potentially related to Curiosity and MIL, i.e. emotional regulation strategies. This capacity labeled Cognitive Reappraisal can support Curiosity in searching for MIL by cognitively reinterpreting emotional experiences important for understanding the meaning of oneself and the world.

Furthermore, we have often drawn attention to the fact that the search for the MIL and the process of solving existential dilemmas is associated with unpleasant emotional experiences. Therefore, we included this variable, i.e. Distress, in our model and we examined its relation to both Curiosity and MIL. We expect there are negative relationships between distress (understood as anxiety and depression) and MIL [44]. Dis-tress, anxiety, and depression can block or inhibit the search for MIL.

Our hypotheses are as follows:

Hypothesis 1: There is the positive predictive role of Curiosity Behaviors along with Flourishing and emotional regulation, and the negative role of Distress (Anxiety, Depression, and Stress) explaining existential vacuum (and the opposite i.e. /Meaning in Life (and its aspects)

Hypothesis 2. Flourishing mediates in the relationship between Curiosity Behaviors and existential emptiness (Meaning in Life and/or its aspects).

Our assumptions are also based on the findings that indicate an impact of emotional regulation and Curiosity on MIL. Likewise, we consider that the state of curiosity is associated with feeling a spectrum of emotions, ranging from positive, including the pleasure of knowledge [37] to negative [15], even aversive [52]. Curiosity exposes individuals to a variety of emotions and, at the same time reinforces the need to explore. It therefore seems to favor strategies based on reinterpretation of the emotional situation. In the process of searching for meaning emotional regulation strategy i.e. cognitive re-appraisal seems to have a positive impact. It can modify one’s search (as well as as-signing) for meaning. In explaining the potential indirect role of emotional regulation strategies we refer to the literature which shows there are negative factors including negative emotional states that decrease MIL and intensify the existential vacuum, and that emotional regulation competencies can modify the process of dealing with negative states while searching for solutions to existential problems [53]. In such states, cognitive reappraisal can be helpful and increase curiosity and MIL.   

Our subsequent hypothesis:

Hypothesis 3. Emotional regulation strategies such as Cognitive Reappraisal and Expressive Suppression mediate in the relationship between Curiosity Behaviors and Meaning in Life (and its aspects).

Our fourth hypothesis is related to the relationship between Curiosity behaviors, Distress, and MIL. The relationships between them can vary. On the one hand, MIL can cause anxiety and distress, and on the other hand, high distress can increase MIL. The correlations between Distress and MIL, and between Curiosity and Distress are negative [44]. Curiosity was found as negatively associated with depression which means that higher Curiosity allows a person to better deal with depressive mood [20]. Likewise, research indicates that low curiosity (high need for cognition) can predict depression symptoms and anticipated anxiety [54]. Sometimes, curiosity can be an emotional negative state [50]. Distress can mediate the relationship between Curiosity and MIL. As we have shown Curiosity may be dependent on Distress and it may modify searching for MIL. Sometimes people experience greater happiness and lower distress and they are more curious and satisfied with their sense of meaning. Another time people are un-happy, distressed, and less curious, and their sense of MIL is lower. During the search for MIL, various interactions between distress and curiosity can occur. These various interactions can even exist in the process of building the meaning: i.e., during the search and the internalization of meaning experiences [45].

Our last hypothesis is as follows:

Hypothesis 4. Distress (including Depression, Anxiety, and Stress) mediates in the relationship between Curiosity Behaviors and Meaning in Life (and its aspects).

Based on the theoretical background and research findings indicating the relationship between Curiosity and MIL, Curiosity and emotion regulation [55], Curiosity and mental/social well-being [56], Curiosity and emotional Distress [57], and relation-ships between MIL, and the all listed variables [27, 51, 53, 58] we developed a model of the potential relationships between these variables (Fig. 1). Taking all these factors together, we aimed to examine what are their associations with MIL, to what extent they explain MIL, and what interactions between them exist that allow explaining existential emptiness.

Reviewer 2 Report

Comments and Suggestions for Authors

You've made a great revision! I have just a few (2) small comments:

Introduction - the beginnings of paragraphs are a bit odd; I would suggest to omit the phrases and just begin with text.

Discussion - it would be better to omit numbering and subsections and just write in plain text.

Author Response

Rev.2

You've made a great revision! I have just a few (2) small comments:

Res. We would like to thank a lot the Reviewer for all helpful comments to the manuscript. All suggestions have been considered.

Introduction - the beginnings of paragraphs are a bit odd; I would suggest to omit the phrases and just begin with text.

Response: OK. We removed it.

Discussion - it would be better to omit numbering and subsections and just write in plain text.

Response: We removed numbering and subsections in the Discussion, the paragraphs are arranged  according to the order of the hypotheses.
